# Computational Analysis of Liquid Hydrogen Storage Tanks for Aircraft Applications

**DOI:** 10.3390/ma16062245

**Published:** 2023-03-10

**Authors:** Vasileios K. Mantzaroudis, Efstathios E. Theotokoglou

**Affiliations:** Department of Mechanics, Laboratory of Testing and Materials, School of Applied Mathematical and Physical Science, National Technical University of Athens, Zografou Campus, 15773 Athens, Greece; v.mantzaroudis@gmail.com

**Keywords:** liquid hydrogen, storage tanks, finite element method, aircraft applications, computational analysis, thermal and structural analysis

## Abstract

During the last two decades, the use of hydrogen (H_2_) as fuel for aircraft applications has been drawing attention; more specifically, its storage in liquid state (LH_2_), which is performed in extreme cryogenic temperatures (−253 °C), is a matter of research. The motivation for this effort is enhanced by the predicted growth of the aviation sector; however, it is estimated that this growth could be sustainable only if the strategies and objectives set by global organizations for the elimination of greenhouse gas emissions during the next decades, such as the European Green Deal, are taken into consideration and, consequently, technologies such as hydrogen fuel are promoted. Regarding LH_2_ in aircraft, substantial effort is required to design, analyze and manufacture suitable tanks for efficient storage. Important tools in this process are computational methods provided by advanced engineering software (CAD/CAE). In the present work, a computational study with the finite element method is performed in order to parametrically analyze proper tanks, examining the effect of the LH_2_ level stored as well as the tank geometric configuration. In the process, the need for powerful numerical models is demonstrated, owing to the highly non-linear dependence on temperature of the involved materials. The present numerical models’ efficiency could be further enhanced by integrating them as part of a total aircraft configuration design loop.

## 1. Introduction

Currently, the most recent forecasts regarding the trends in the aviation sector’s evolution predict a rapid growth during the next 20 years [1,2]. The most challenging part of this growth deals with the replacement of the ageing parts of the global fleet as well as with the production of an additional number of aircraft in order to meet the increasing passenger and cargo transport needs; the total number reaches a level of 40,000 new aircraft. The management of this issue becomes even more challenging when combined with the remarkably high level of global greenhouse gas emissions, which has led global organizations to decisions and actions for not only restraining but totally eliminating the emissions in the coming decades, e.g., the European Green Deal [3]. It is noted, though, that one-sided investments in the improvement of the current fossil-fuel technologies are not enough for achieving this target, and consequently, the funding of innovative green technologies arises as a high-potential solution [4].

In this context, a technology that has already been known to the scientific and industrial community for decades now, and has been tested to a certain extent but mainly at the level of small-scale experimental attempts or at aircraft prototype test flights and not on fully operational aircraft, is the use of hydrogen (H_2_) as fuel for the ever-growing global aircraft fleet. Among the different methods of hydrogen storage, an efficient one that is applicable to aircraft structures is cryogenic storage in the liquid state (LH_2_) at very low temperatures (−253 °C) and relatively low pressures, significantly lower than those required for the storage of H_2_ in its gaseous state (GH_2_). According to Table 1 [4,5,6], LH_2_ possesses almost three times (≈2.8) the energy content per unit mass (J/kg) compared to kerosene, which means that for two engines with the same specific consumption, the one using H_2_ as fuel will achieve an almost three times larger range compared to a kerosene-fuelled corresponding one. However, due to the significantly lower energy content per unit volume of LH_2_ (≈four times), a challenging matter in the management of LH_2_ storage in aircraft deals with balancing the different effects that the high energy content per unit mass and per unit volume have. More specifically, matters of lower fuel mass, which positively affect the total aircraft mass, come in contrast with matters of increased LH_2_ tank volume and, consequently, aircraft configuration volume, which negatively affects the aerodynamic behavior through the lift-to-drag ratio (L/D).

The idea of incorporating H_2_ as fuel in full-scale aircraft applications, stored as LH_2_, is not recent, with the first ever occasion regarding the B-57military aircraft of the US [7,8] in the 1950s, where the existing propulsive system was modified to incorporate an engine running on both kerosene and H_2_. A joint venture between Airbus and the US Space Agency (NASA) in the 1990s conducted modification studies on the existing Airbus A310 aircraft, resulting in three design proposals for the integration of LH_2_ tanks in the aircraft structure [7,9]. A more recent project by the Boeing Company in the early 2010s involved the development of the Phantom Eye UAV, which incorporated a spherical LH_2_ tank [7].

Furthermore, a remarkable amount of research has been produced within the last 20 years that concerns the design of innovative or the modification of existing aircraft to use hydrogen stored as LH_2_ in suitable tanks. In the 2000s, the European Union launched the CRYOPLANE project [9,10,11], where a consortium of 35 industrial and academic partners undertook studies on the prospect of designing aircraft configurations for hydrogen propulsion, storing it as LH_2_. In addition, a number of investigations developed the design issues of integrating LH_2_ tanks into aircraft structures. Characteristic case studies in conventional aircraft are those of Verstraete et al. [12], Silberhorn et al. [13] and Onorato [14]. Even more, there are studies that concern unconventional aircraft configurations, with the most characteristic ones being those of Seeckt et al. ([15], Project Green Freighter, Blended Wing Body (BWB) configuration), Smith ([16], BWB configuration), AHEAD project ([17], BWB configuration), Goldberg ([18], BWB configuration), Dietl et al. ([19], Forward Swept Wing configuration) and van Woensel ([20], Flying-V aircraft concept). A common feature of these studies, however, is that they have involved the LH_2_ tank design in the overall design loop of the aircraft, based on analytical–semi-analytical–empirical literature solutions, without developing solutions by incorporating computational methods.

Finally, a number of studies concerning thermal and structural issues of LH_2_ tanks exist in the literature. Tapeinos et al. [21,22,23] carried out a computational and experimental study of an LH_2_ tank in a multiple interconnected sphere configuration made of a polymer and composite double shell separated by a layer of insulating foam; the tank was studied in various temperature and pressure scenarios, while issues of reinforcing structural elements in critical areas of the tank outer surface were investigated as well. Gomez and Smith [24], based on the MRT7-3 “Meridian” medium-range aircraft design developed by Cranfield University, performed a finite element computational study for the installation of proper LH_2_ tanks, including modifications of the initial airframe and taking into account the temperature and pressure at the tank interior.

From the literature it appears that there is a lack of computational analyses dealing with tanks of different geometry configuration and LH_2_ level stored. In the present work a static computational analysis is performed via the finite element method in order to parametrically analyze such tanks, examining the effect of the level of LH_2_ stored and the tanks’ geometric configuration on their thermal and structural performance, taking under consideration the temperature-dependent properties of the materials involved. Consequently, a set of guidelines for LH_2_ tank dimensioning are formulated, useful as a design tool in the decision-making procedure when integrating them in an aircraft configuration. In the process, the need for powerful numerical models is demonstrated, owing to the involved materials’ highly non-linear dependence on temperature. The efficiency of the present numerical models could be enhanced by implementing them as part of a total aircraft configuration design loop.

## 2. Materials and Methods

In this section, a detailed description of the problem of LH_2_ storage tanks and the methods by which the analyses were performed is carried out. For the implementation of the objectives studied and the assumptions [(a)–(d)] considered, the following are defined:(a)The tank configuration is selected to consist of the following three (3) structural components:
An inner shell that is in touch with the LH_2_ (commonly mentioned in the literature as “liner”), made of an aluminum alloy. The reason a metallic shell is selected, compared to the other material category that could possibly be incorporated, that of composite materials, is that metallic materials exhibit a more efficient behavior in terms of hydrogen diffusion through their microstructure and of property degradation due to embrittlement when in contact with hydrogen. The possibility of using a combination of metallic and composite shell was not studied, due to the danger for catastrophic failure arising from the different stress field in their interface, which is a result of the much different coefficients of thermal expansion (CTEs) that these two material categories exhibit.Insulating material made of polyurethane (PUR) foam, which is characterized as efficient and well-established in terms of knowledge of its mechanical and thermal properties and its general behavior, as well as with low cost and satisfactory thermal conductivity [25].An outer shell made of layers of composite material to save mass, since this outer shell of composite material mainly plays the role of protecting the insulating material from external loads that are likely to cause a catastrophic failure.(b)The tanks under consideration are chosen to belong to the “non-integral” category, i.e., not to be part of the aircraft structure but to function simply as LH_2_ storage containers (pressure vessels). The reason for this choice is that the special characteristics of the thermal and structural behavior of a large variety of tank configurations could be highlighted, regardless of any specific aircraft design in which they are intended for use. Therefore, two additional issues, which are clarified below, are mentioned:
The shape of the considered tanks. To produce all of the tank geometries considered, the method of triaxially symmetric tank geometries followed by Winnefeld et al. [5] was adopted. According to it, three (3) characteristic tank size ratios, λ, φ and ψ, are enough to describe its shape, as shown in Figure 1. These ratios are connected to the dimensions a, b and c of Figure 1, which are emphasized as being the semi-axes of three (3) ellipses (a–b, a–c, b–c), which, together with the length of the central part of the tank, l_s_, completely describe the tank.

At this point it is necessary to point out that the schematic description of Figure 1, for the purposes of this work, expresses the geometry of the inner aluminum shell (liner) in which the stored LH_2_ is enclosed. The thickness of the insulating material is an object of study and is placed outside the geometric shape described from Figure 1. Additionally, it is emphasized that in all analyses carried out in the present study, the volume enclosed by the inner aluminum shell is an object of choice. That is, for each tank under consideration, the volume enclosed by the inner aluminum liner, V_total_, as well as the values of the three (3) geometric ratios λ, φ and ψ (Figure 1), are initially selected. Finally, since it is common practice not to completely fill this volume with LH_2_, but instead the maximum percentage that is allowed is on the order of 93% [26], the height f expressing the level of liquid hydrogen is calculated iteratively (with continuous changes in its value until the desired volume is reached).
 •The temperature and pressure conditions that apply inside and outside the tank and constitute the thermal and structural loads of the tanks under investigation, as “non-integral”, are the only ones they undergo (in contrast to the integral ones, they do not undergo the aerodynamic loads that the aircraft structure (fuselage and wings) receives). The temperature of the hydrogen is therefore taken to be equal to −253 °C (20 K), while the pressure in the volume enclosed by the inner shell is equal to 170 kPa [27]. The temperature of the external environment is taken, indicatively and by no means limiting, as equal to 10 °C (283 K).

(c)As far as the thermal aspect of the issue is concerned, heat transfer between the external tank environment and the hydrogen enclosed in the inner tank shell is governed by all three (3) forms of heat transfer:
Heat transfer between the external environment and the composite shell due to radiation.Heat transfer between the external environment and the composite material shell due to convection. For the purposes of this work, a value of 10 W/m^2^ K was taken, within the typical range of convection coefficients for free air flow (2.5–25 W/m^2^ K, [28]).Heat transfer between the composite shell and the insulating material, as well as between the insulating material and the aluminum shell due to conduction.Heat transfer between the aluminum shell and the hydrogen, either in its liquid or gaseous form, due to convection. In this case, it is necessary to determine two different convection coefficients, one for gaseous hydrogen and one for liquid, the values of which are determined as follows [29]:


Hydrogen gas: 1.4587 W/m^2^ KLiquid hydrogen: 2.1827 W/m^2^ K

(d)Regarding the structural aspect, in addition to the 170 kPa pressure previously mentioned to exist inside the inner aluminum shell, additional stress is created by the temperature distribution through all the tank components. This distribution is expected to show large variations, since, as mentioned above, the temperature in the tank interior is equal to −253 °C (20 K), while the external environment was considered to be 10 °C (283 K). These temperatures, through the coefficients of thermal expansion (CTEs) of the tank component materials, create thermal stresses that are added to the mechanical stresses due to pressure in the tank interior. On the matter of possible tank structural supports, it is remarked that the tanks under investigation are not considered to be installed in a specific aircraft structure, but instead they are supposed to be simply resting on the ground, without suppressing the displacement of any specific part of it, in order to exhibit the unique characteristics of each tank configuration, unaffected by possible stress concentrations or other influence due to complex support conditions.(e)In regard to the expression of the criteria according to which the thermal and structural behaviors of the tanks are assessed:
For the thermal behavior, the achievement of a specific vaporization rate of the liquid hydrogen (boil-off rate) is defined as a criterion, the desired value of which is chosen before solving the problem. Through the following Equation (1) [29], this vaporization rate is connected to the maximum allowed heat flow between the hydrogen and the external tank environment.

(1)$$BOR\;\left(Boil-off\;rate\right)=\frac{360,000\cdot Q}{\Delta h\cdot m_{LH_{2}}}=\frac{360,000\cdot Q}{\Delta h\cdot V_{LH_{2}}\cdot \rho_{LH_{2}}}\left(\%/h\right)$$
where:Q is the total heat flow between the LH_2_ inside the tank and the external environment (W)Δh is the heat of vaporization (J/kg), and its value for LH_2_ is taken to be equal to 461,000 J/kg [4,5,6].$m_{LH}{}_{{}_{2}}$ is the mass of LH_2_ inside the tank, which is calculated as the product of the LH_2_ volume (determined by the tank inner shell dimensions and the percentage of fullness of the tank) and its density, $\rho_{LH}{}_{{}_{2}}$ (=71 kg/m^3^) (Table 1).
For the structural behavior, a separate criterion is defined for each of the tank components:


For the aluminum inner shell, the Von Mises stress criterion is selected, with the critical temperature-dependent value obtained from experimental results published in the literature, as designated in the next paragraphs.For the PUR foam insulation material, the Tsai–Wu index criterion is chosen, which is built into the library of the software used to perform the analyses, and which has been demonstrated to be valid for the case of a corresponding type of foam (polyvinylchloride, PVC) [30].For the composite material outer shell, two (2) different criteria from those available in the software [31] are selected, the Tsai–Wu and the Hashin ones.

Concluding, the computational analysis process for each tank is the following:(a)Calculation, through repeated iterative analyses, of the insulation thickness required to achieve the tank heat flow rate dictated by the selected hydrogen vaporization rate (boil-off rate), by performing thermal analysis, taking into account the heat transfer modes mentioned above, as well as the temperature conditions inside and outside the tank as boundary conditions.(b)Saving the temperature distribution calculated during the solution of the thermal analysis at all nodes of the tank and introducing them as boundary conditions in the structural analysis, along with the pressure condition of 170 kPa inside the tank.(c)Keeping the thickness of the insulating material and the composite material outer shell constant, calculation through iterative analyses of the thickness of the aluminum inner shell until the Von Mises criterion is satisfied, which acts as an evaluation criterion for the aluminum alloy. At the same time, confirmation that both the insulating material and composite shell evaluation criteria do not indicate any failure in these tank components.

The development of the numerical models used to perform the required analyses of the present study, which were built in the APDL© (ANSYS Parametric Design Language) [31] software environment, is followingly presented.

For both thermal and structural analyses, the tank components described previously are modeled with three-dimensional (3D) 8-node finite elements, which are the SOLID278 for the thermal part of the analysis and the SOLID185 for the structural part. Regarding the SOLID278 element, it employs 1 degree of freedom (DOF) in each node, which is the nodal temperature, while the SOLID185 employs 3 DOFs in each node, which are the nodal translational displacements in each one of the x, y and z element axes. Both elements come in two versions in the APDL© environment:The homogeneous version, where the element is considered to be of isotropic behavior and, as such, is used to model isotropic materials.The layered version, where the element is internally divided in multiple layers, and by selecting a different for each layer axes system, anisotropic materials are modelled.

The homogeneous and layered version of the SOLID278 and SOLID185 elements are depicted in Figure 2.

For the case of the problem under study, the material models used were all linear, since the investigation is focused on the study of the elastic region of the problem. The “non-linearity” of the phenomena in the specific problem results from the dependence of the thermal and mechanical properties of the materials on the temperature, as has been analyzed and shown through experimental results published in the literature [5,6,7,8,9,10,11,12,13,14].

For the inner shell metallic material, the aluminum alloy considered is Al 2219-T87, which is a proven and well-characterized alloy for cryogenic applications (see Brewer [27]). For the problem studied in this work, the required physical, thermal and mechanical properties of the specific aluminum alloy are taken from the study of Simon et al. [32]. Regarding the insulating material layer, polyurethane (PUR) foam of density 64 kg/m^3^ was considered. The experimental results for the characterization of the foams in terms of their thermal and mechanical properties were taken from the work of Sparks et al. [33]. For modeling the outer composite shell, it is important to mention that, given the nature of the problem, the shell is expected to be very close to the ambient temperature chosen, i.e., 10 °C (283 K). Therefore, knowledge of the cryogenic properties of the composite material is not required. For the present study, a composite material of polymer matrix with carbon fibers (Composite Fiber Reinforced PolymerCFRP) of the company Hexcel© was considered, namely AS4/8552, transversely isotropic, the required thermal and mechanical properties of a single layer (ply) of which are included in Table 2 [21,22,23]. The stacking sequence of the composite shell was chosen to be a quasi-isotropic one, namely [0/45/90/-45]_s_, in order to avoid the involvement of load-displacement coupling phenomena that are inherent to other stacking sequences. Different materials may also be considered for the composite shell [34,35].

## 3. Results

### 3.1. Convergence Study

In order to select the appropriate mesh size to perform the structural and thermal analyzes of the tanks under consideration, a short convergence study is first carried out, selecting one case of the tank geometries described in Figure 1, with geometric ratios: λ = 0.5, φ = 1, ψ = 1. In addition, as a reference point for determining the values of the geometric dimensions of the tank based on the above three ratios, the volume of the liquid hydrogen storage space is taken to be equal to 100 m^3^. Furthermore, the thickness of the insulation foam placed between the aluminum shell and the composite material shell is specified, equal to 0.6 m. Schematically, the considered tank arrangement is illustrated in Figure 3.

For an examination of the tank mesh convergence, different mesh sizes regarding both the in-plane size of the elements and the through-thickness one were chosen (Figure 4). More specifically:Regarding the in-plane mesh size (length and width)
○0.15 m (coarse mesh)○0.1 m (medium mesh)○0.05 m (fine mesh)Regarding the through-thickness mesh size
○For the elements of the aluminum inner shell and the composite outer shell: one element (due to small thickness)○For the elements of the insulation polyurethane foam layer:
▪four elements through thickness (coarse mesh)▪six elements through thickness (medium mesh)▪eight elements through thickness (fine mesh)

To control the resulting different mesh densities, the following quantities are considered: (a) The thermal output of the tank; (b) the maximum and minimum temperature overall in the tank; (c) the distribution of temperatures (temperature profile) in three points of the tank, focusing on the part of the insulation where the main part of the change in temperatures between the inside and the outside of the tank occurs (points of interest depicted in Figure 5). Following the above, Table 3 is made, where the results related to the aforementioned elements are recorded, as well as Figure 6, Figure 7 and Figure 8, where the temperature distribution through the insulation thickness at the three different points of Figure 5 is depicted.

From Table 3 and Figure 6, Figure 7 and Figure 8, we have:(1)For the values of the minimum and maximum temperature overall in the tank, it is observed that only a negligible effect, if not zero, causes both the change in the dimensions of the elements in the plane and the number of elements in the thickness of the insulation.(2)For the size of the heat output, Q, of the tank:
By keeping the number of elements constant across the thickness of the insulation, dimension variation of the elements has no significant influence on Q.On the contrary, by keeping the dimensions of the elements constant and varying their number by the thickness of the insulation, it is found that the changes are more significant than in sub-case (a).

It is therefore deduced that the choice of the number of elements per thickness seems to play an important role in the distribution of temperatures in approximately one-third of the thickness of the insulation that is closest to the hydrogen storage area. The same situation is observed almost identically at all three points chosen to obtain the distribution of temperatures along the thickness of the insulation. Therefore, for the discretization of the reservoirs, in the finite element analyses that follow, the “medium discretization” is used in terms of the dimensions of the elements in the plane, while for the thickness, the number of eight elements is chosen, exclusively for the reason of assurance that in the event of a greater thickness of insulating material, the number of elements will be sufficient for the accurate depiction of the temperature profile.

### 3.2. Study of the Tanks Relative to the Height of LH_2_

As already mentioned, the maximum LH_2_ filling percentage of the volume enclosed in the inner tank shell is of a level of 93% [26]. The remaining space is occupied by gaseous hydrogen (GH_2_). Regarding the performed numerical analyses, the LH_2_ tank fullness percentage affects the convection coefficient in the interior of the tank, with the values for GH_2_ and LH_2_ already given in Section 2. At this point, and for the same form of tank considered in the convergence study presented above (V_total_ = 100 m^3^, λ = 0.5, φ = 1, ψ = 1), a thermal and structural analysis is performed for a tank filled with LH_2_ at the maximum percentage of 93%. In the following Figure 9, Figure 10, Figure 11 and Figure 12, the depiction of the following results is included:Temperature distribution for the three tank components (inner aluminum shell, insulating material, outer composite shell) (Figure 9).Heat flux (heat flow per unit area) for the three tank components (inner aluminum shell, insulating material, outer aluminum shell) (Figure 10).Von Mises stress on the inner aluminum shell (Figure 11a).Tsai–Wu index on the insulating material (Figure 11b).Tsai–Wu index on the outer composite material shell (Figure 12a).Hashin fibre failure criterion on the outer composite material shell (Figure 12b).Hashin matrix failure criterion on the outer composite material shell (Figure 12c).

From the analyses it is observed that, in the case of the aluminum shell, two distinct temperature regions are created: the region of the shell in contact with GH_2_, where the calculated temperatures are in the range of about 25 to about 27 K, and the region of the shell in contact with LH_2_, where the calculated temperatures are in the range of about 23 K to about 25 K. In the case of the insulating material, due to the very large change in temperatures from the side in contact with the aluminum shell to the side in contact with the composite material shell (in total, a change from about 23 K to about 282 K), no significant difference in the temperature distribution across the thickness of the insulation is depicted. Finally, the outer composite material shell practically does not undergo any significant temperature change, since the difference between its two surfaces is only 0.1 K, and it is not affected by the percentage of fullness of the tank.

Regarding the distribution of the heat flow per unit surface (heat flux), it is observed that a thin region of significantly increased heat flux appeared in the aluminum shell at the gas–liquid hydrogen interface, two orders of magnitude above the flux observed across the rest of the surface. In the insulating material, the heat flux varies between 5.18 and 10.58 W/m^2^, with no noticeable correlation with the height of LH_2_ in the tank. In the composite shell, the heat flux varies consistently between 5.13 and 5.97 W/m^2^; however, the area with the maximum value of 5.97 W/m^2^ appears to correlate with the LH_2_ level, and more specifically it is located in the area of the cylindrical body of the tank that is in contact with GH_2_.

### 3.3. Study of the Tanks under Constant Volume of LH_2_

In this section a set of different tank designs are examined, which result from different values of the geometric ratios λ, φ and ψ, which have already been explained in Section 2. In order to be able to compare the behavior of the different tank configurations, a common volume of the inner tank shell is selected for all of them, equal to 100 m^3^. By the three ratios λ, φ and ψ, and the volume of 100 m^3^, the characteristic dimensions a, b, c, l_s_ and l_t_ mentioned in Section 2 are calculated.

The objective here is to study the different tank configurations from a thermal and structural point of view in order to determine the required tank component thicknesses (inner shell, insulating material, outer shell). As criteria for the definition of the above, the following considerations are made:For the thermal behavior of the tank, the LH_2_ vaporization rate (boil-off rate) is taken as a criterion, which is a direct function of the heat flow between the LH_2_ and the tank external environment, based in Equation (1). The boil-off rate chosen in this particular case is equal to 0.1%/h, based on those mentioned by Mital et al. [33] for the case of aviation applications. Therefore, considering for the tanks of this Section an LH_2_ filling percentage of 93% (the maximum limit set, as already mentioned in Section 2), and a volume of the inner shell of 100 m^3^, the heat flow maximum allowable limit to achieve the selected boil-off rate is calculated equal to 845 W.For the structural behavior of the tank, as previously, the Von Mises stress criterion for the inner aluminum shell, the Tsai–Wu index criterion for the insulating PUR foam and both the Tsai–Wu index and Hashin criteria for the outer composite material shell are used.

Consequently, the developed method of this section consists of:Iterative thermal analysis of the tank in order to determine the insulating material thickness required to marginally achieve the heat flow limit of 845 W (the insulating material thickness is taken as constant in every point of the tank).Iterative structural analysis of the tank, by incorporating the temperature distribution extracted by the thermal analysis in order to create the thermal stresses and additionally applying an internal pressure of 170 kPa, in order to determine the inner aluminum shell thickness that marginally achieves the Von Mises stress criterion. In parallel, the insulating PUR foam and the outer composite material shell are also evaluated for their structural integrity by means of the structural criteria mentioned above. It is noted that, in order to avoid complexity matters, the thickness of the outer composite shell is kept constant, with the value defined in Section 2, and it is only evaluated against failure. Table 4 includes details of the tank configurations considered in this section.

For a visual insight of the tank shapes, and by means of the tank views shown illustrated in Figure 13, Figure 14, Figure 15 and Figure 16 depict the typical shapes resulting from the values of Table 4 for the case of ratio λ = 0.25, with similar shapes occurring for the other two ratios, λ = 0.5 and 0.75.

Diagrams are constructed afterwards for the tank cases of Table 4 in commonly calibrated axes, which include the depiction of the following results extracted by the thermal and structural analyses and based on which the assessment of their behavior is performed. The diagrams of Figure 17, Figure 18, Figure 19, Figure 20, Figure 21 and Figure 22 include:The total surface of the inner aluminum shell based on the characteristic dimensions l_s_, l_t_, a, b and c. (Figure 17).The necessary insulating PUR foam thickness in order to satisfy the requirement for a boil-off rate with a value of 0.1%/h from which the maximum allowed heat flow between the tank interior and the external environment is calculated equal to 845 W (Figure 18).The total mass of the insulating material, which is affected by both the total surface of the inner aluminum shell and the insulating material thickness (Figure 19).The thickness of the inner aluminum shell in order to satisfy the Von Mises stress criterion, according to which its structural integrity is assessed (Figure 20).The total tank mass (inner shell, insulating material, outer shell) (Figure 21).The gravimetric efficiency of the tank, which is defined by Equation (2) and is a general means of demonstrating the tank efficiency in storing the LH_2_ in terms of tank mass [12] (Figure 22):
(2)$$n_{grav}=\frac{m_{LH_{2}}}{m_{LH_{2}}+m_{\tan k}}=\frac{W_{LH_{2}}}{W_{LH_{2}}+W_{\tan k}}\;$$

Observing the diagrams of Figure 17, Figure 18, Figure 19, Figure 20, Figure 21 and Figure 22, the following conclusions arise:Regarding the minimum insulating material thickness that is calculated in order to maintain the LH_2_ vaporization rate (boil-off rate) at a level of 0.1%/h (Figure 18), as it has been defined, it is concluded that, in general, the required thickness is connected to the surface of the inner aluminum shell. More specifically, the larger the surface, the thicker the insulating material that needs to be applied in order to ensure the proper heat flow that corresponds to the specified boil-off rate of 0.1%/h. This phenomenon is generally expected, because the heat transfer mechanism of convection occurring between the hydrogen in the tank interior and the inner aluminum shell is dependent on the total surface of solid–fluid interaction. Exceptions to this trend are the tanks with geometric ratios λ = 0.75, φ = 0.333, ψ = 1.5 and λ = 0.75, φ = 0.333, ψ = 3, for which it is noted that the insulating material thickness is kept almost steady compared to the previous one (λ = 0.75, φ = 0.333, ψ = 1). This is attributed to the fact that for the specific two tanks the required insulation thickness that ensures the proper boil-off rate is very close to the critical insulation thickness beyond which any further increase would also increase, instead of reducing, the heat flow; this phenomenon is attributed to the fact that the outer composite shell surface is increased so much due to the increased insulation thickness that results in an increased heat flow due to convection between the outer shell and the external environment.As for the insulating material mass, which is calculated according to the already-calculated insulation thickness (Figure 19), it is observed that the distribution is strictly following the corresponding one of the inner aluminum shell surface, as expected. In this case as well, exceptions are the two tank configurations (λ = 0.75, φ = 0.333, ψ = 1.5 and λ = 0.75, φ = 0.333, ψ = 3) mentioned, where the distribution slope is affected by the straightening of the thickness distribution, which was explained in the previous paragraph.Concerning the required thickness of the inner aluminum shell to satisfy the Von Mises criterion (Figure 20), based on which its structural strength is evaluated, it is observed that this does not show any specific pattern between the different tanks. This is due to the fact that overall the arrangement of the examined tanks is such that their structural behavior is quite complex, and certainly non-linear, due to the fact that the thermal and mechanical properties of the materials used are a function of temperature. Specifically, characteristic points of the diagrams that are worth highlighting, and concern areas with a sharp change in the distribution between the different configurations, e.g., for the tanks with λ = 0.25 and φ = 0.333: during the transition from ψ = 1 to ψ = 1.5 (Figure 20), a sharp change in the distribution is observed, which is due to the fact that the point where the maximum Von Mises stress is observed (the most critical area of the aluminum shell) shifts from one point of the tank to another. A representative depiction of this phenomenon is included in Figure 23.Concerning the total mass of the tank (Figure 21), it is already clear from the above that it follows the distribution of the mass of the insulating material, since this constitutes the overwhelmingly greater percentage of the total mass of the tanks, since in all cases this percentage is within the range 87–95%.Finally, regarding the gravimetric efficiency of the tanks (Figure 22), due to the domination of the insulating material mass, it follows a reverse distribution and exhibits a range of values between 15 and 54%.

## 4. Discussion and Results

The tank fullness level, at least for the issues considered in the present study, does not seem to substantially affect the two parts of the tank into which the liquid hydrogen level separates very differently. A small difference in the temperatures of the aluminum shell, on the order of 2 °C (2 K), is observed due to its direct contact with liquid and gaseous hydrogen, while the same temperature difference is observed in the lower layer of the insulating material, which comes in contact with the aluminum shell; however, it is quickly annihilated by the thickness of the insulation layer, due to its low thermal conductivity. Additionally, a thin band of increased heat flux is observed at the level of LH_2_. Both of these phenomena are due to the different convection coefficients for GH_2_ and LH_2_. In terms of structural behavior, this separation of the reservoirs into two areas creates a difference in the values of the failure criteria, with the area in contact with LH_2_ being more sensitive due to the lower temperature that it is forced to reach.

For a given tank volume, it becomes clear that the thickness and corresponding mass of the insulating material generally depends on the total surface area of the LH_2_ storage chamber, since heat flow effects are a function of the exposed surface area. This means that, with a selected volume of LH_2_ desired to be stored, only by knowing the geometric arrangement of the tank through the chosen ratios λ, φ and ψ is it possible to form an understanding of the comparative thickness of insulation between different arrangements. However, it is of great importance to consider the temperature dependence of the mechanical and thermal properties of the materials involved; especially for the insulating material, which undergoes the most severe temperature gradients due to its very low thermal conductivity, only by means of powerful computational models are meaningful results possible to be obtained. Regarding the required thickness and corresponding mass of the inner aluminum shell, they are dependent on the tank geometry, with tanks possessing quite large or small curvatures being very sensitive to the Von Mises stress distribution, which is the evaluation criterion, with a high risk of displacing the point of this stress maximum value to a totally different area of the tank.

The calculations performed in the present work, where emphasis was placed on incorporating the temperature-dependent non-linearities and examining a large variety of tank shapes, guide the engineer to evaluate and select tanks in order to fit a specific LH_2_ volume in an aircraft structure by determining the final outer dimensions of the tank, which is proven useful from a design point of view in the early stages of aircraft design incorporating LH_2_ tanks. Consequently, the use of new storage tanks for temperature-dependent non-linearities and different tank geometries have a great influence in new airplane structures. In addition, liquid hydrogen tanks, in addition to the aviation industry, may be deployed in maritime transport industries, industrial chemistry, etc. The proposed study concerns a global thermal and static computational analysis for parameterized simplistic tank geometries; the case of dynamic analysis, different material properties and the introduction of governing equations related to a specific geometry with large or small curvatures will be the purpose of a future study.

## Figures and Tables

**Figure 1 materials-16-02245-f001:**
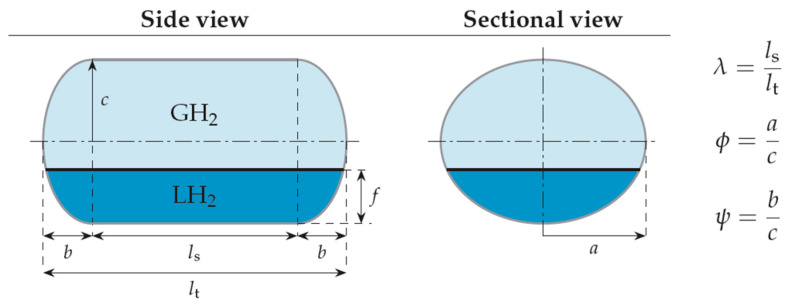
Production of geometric shapes of tanks through three geometric ratios of their main characteristic dimensions [5].

**Figure 2 materials-16-02245-f002:**
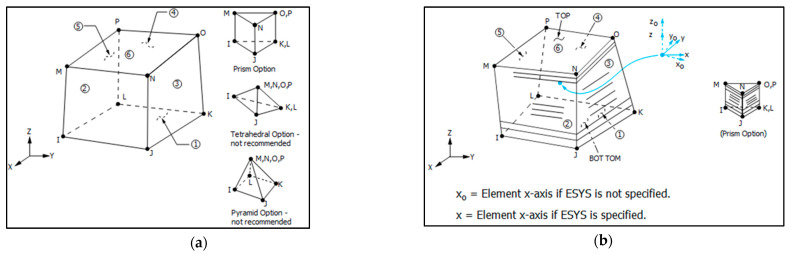
Graphical representation of the homogeneous (**a**) and the layered (**b**) version of SOLID278 and SOLID185 elements of the APDL© software [31].

**Figure 3 materials-16-02245-f003:**
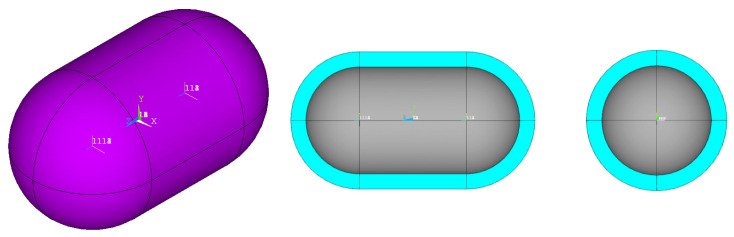
Representation of the tank configuration on which the convergence analysis is performed.

**Figure 4 materials-16-02245-f004:**
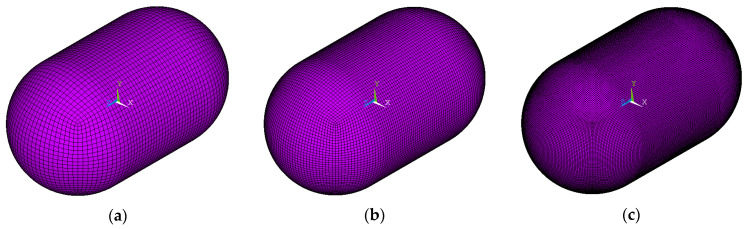
General view of the tank with the different mesh sizes examined: (**a**) coarse, (**b**) medium, (**c**) fine.

**Figure 5 materials-16-02245-f005:**
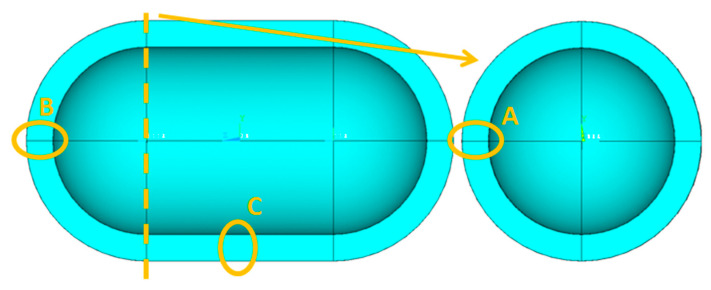
Points A, B, C of interest for temperature distribution calculation through the insulation thickness.

**Figure 6 materials-16-02245-f006:**
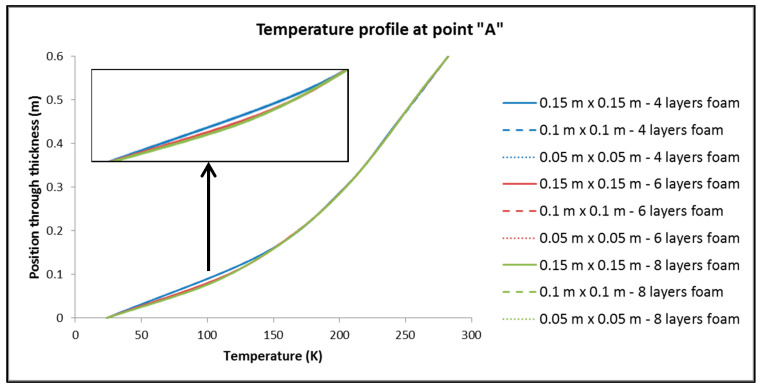
Temperature profile through the insulation thickness at point “A” of Figure 5.

**Figure 7 materials-16-02245-f007:**
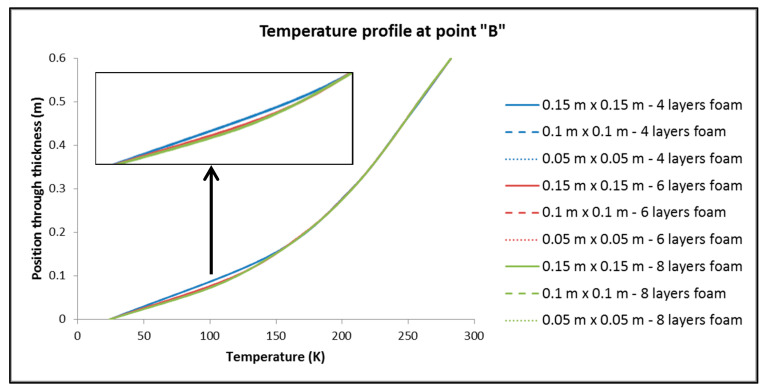
Temperature profile through the insulation thickness at point “B” of Figure 5.

**Figure 8 materials-16-02245-f008:**
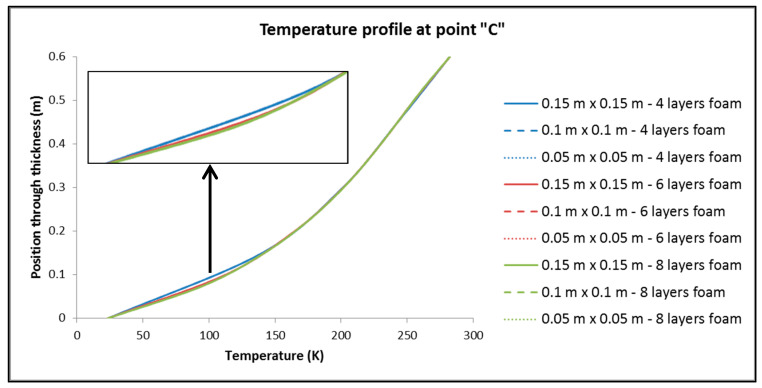
Temperature profile through the insulation thickness at point “C” of Figure 5.

**Figure 9 materials-16-02245-f009:**
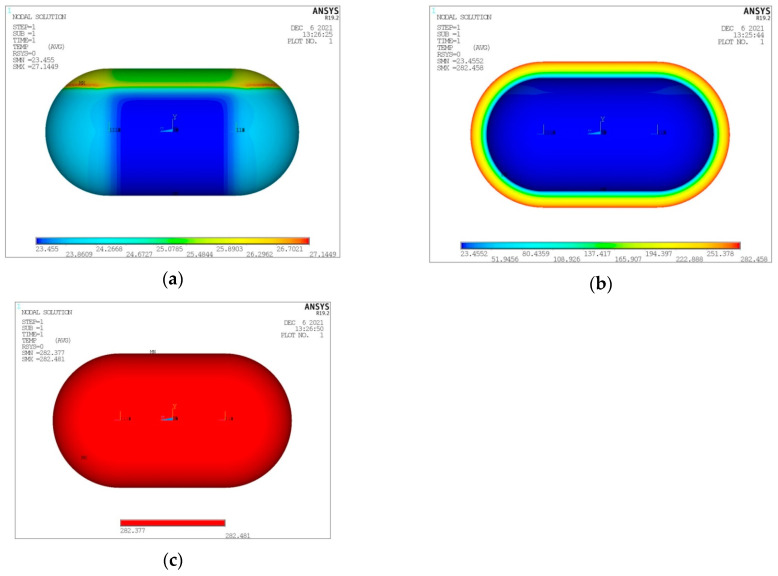
Temperature distribution for (**a**) inner aluminum shell, (**b**) insulating material, (**c**) outer composite shell.

**Figure 10 materials-16-02245-f010:**
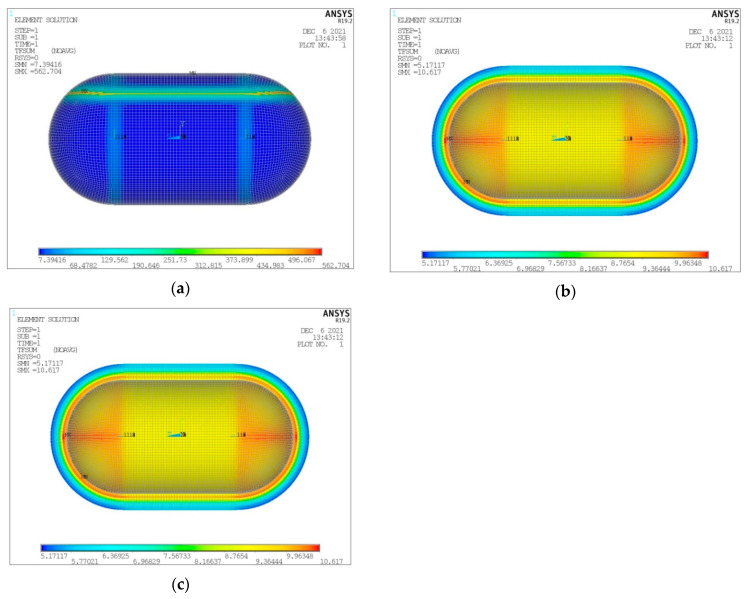
Heat flux distribution for (**a**) inner aluminum shell, (**b**) insulating material, (**c**) outer composite shell.

**Figure 11 materials-16-02245-f011:**
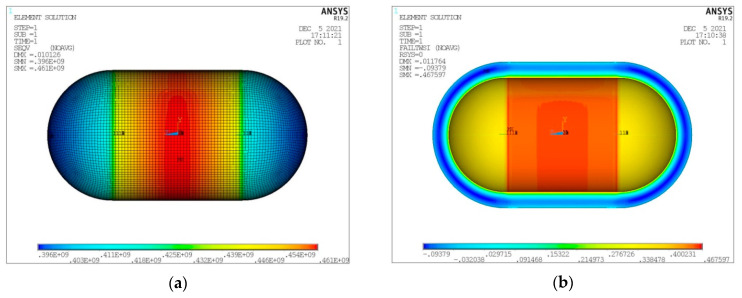
Distribution of (**a**) Von Mises stress on the inner aluminum shell and (**b**) Tsai–Wu index on the insulating material.

**Figure 12 materials-16-02245-f012:**
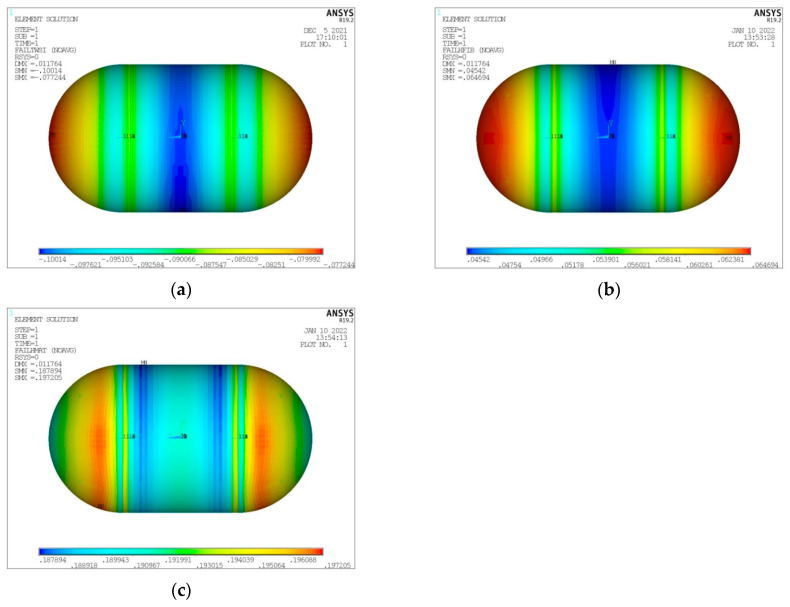
Distribution of (**a**) Tsai–Wu index, (**b**) Hashin fiber failure criterion and (**c**) Hashin matrix failure criterion on the outer composite shell.

**Figure 13 materials-16-02245-f013:**
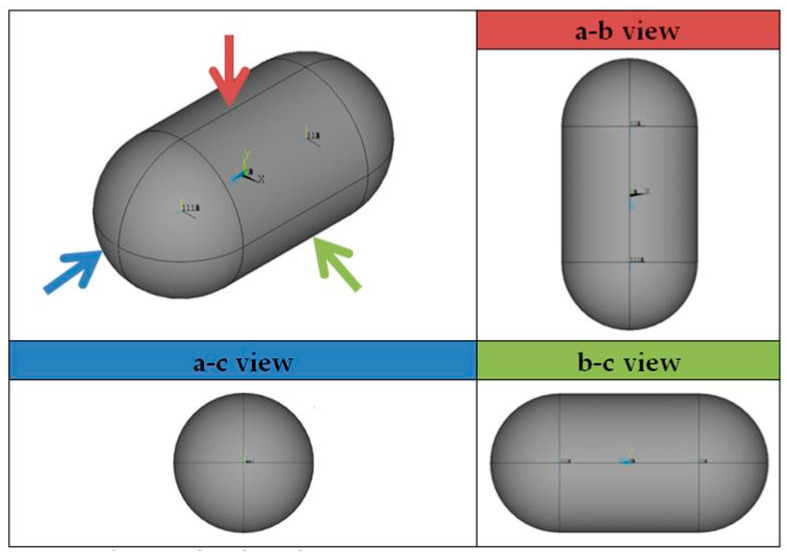
Views of a typical tank configuration.

**Figure 14 materials-16-02245-f014:**
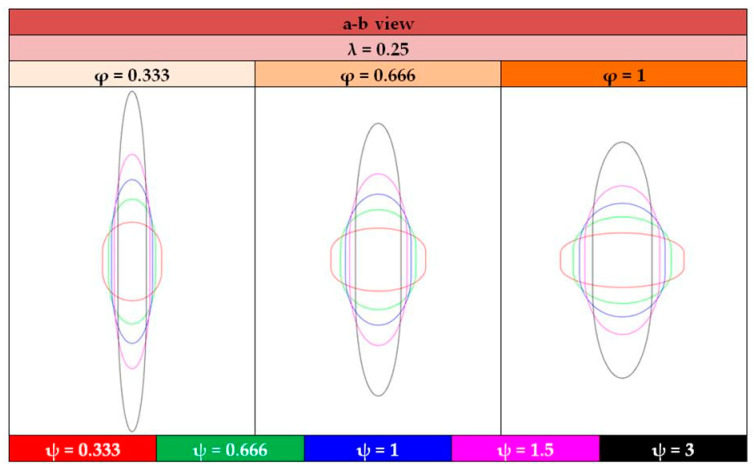
Graphical representation of the a–b view of tank configurations with λ = 0.25.

**Figure 15 materials-16-02245-f015:**
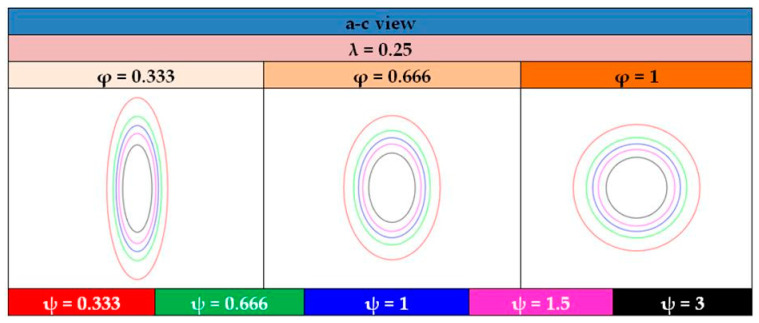
Graphical representation of the a–c view of tank configurations with λ = 0.25.

**Figure 16 materials-16-02245-f016:**
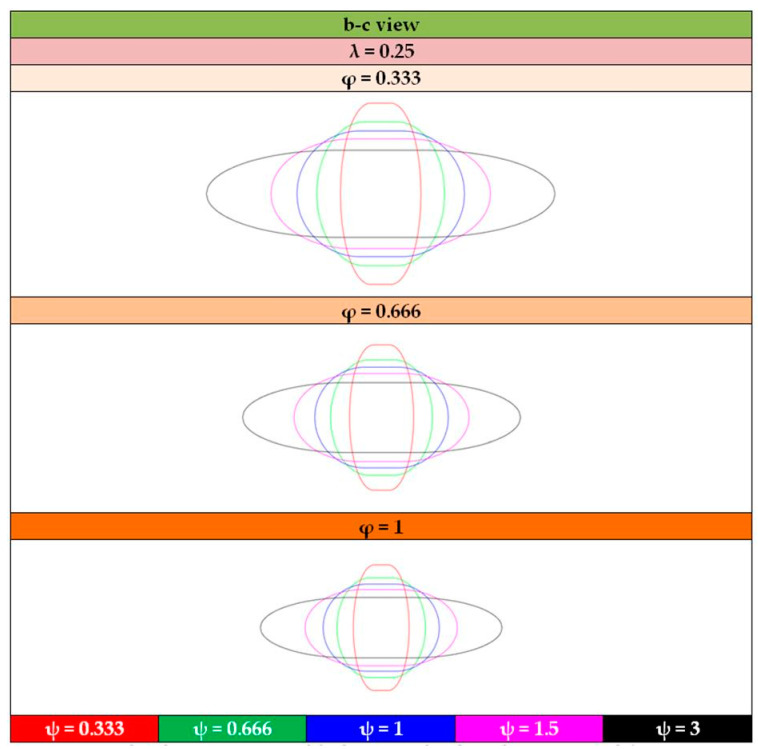
Graphical representation of the b–c view of tank configurations with λ = 0.25.

**Figure 17 materials-16-02245-f017:**
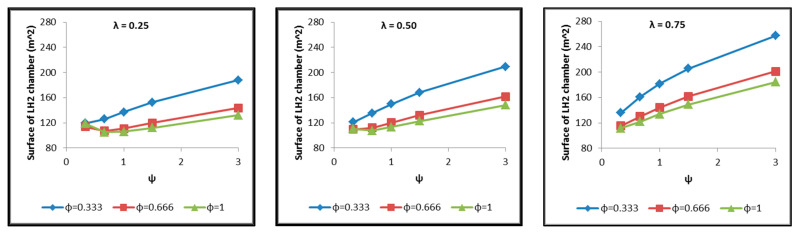
Distribution of the inner aluminum shell surface as a function of the ratios λ, φ and ψ.

**Figure 18 materials-16-02245-f018:**
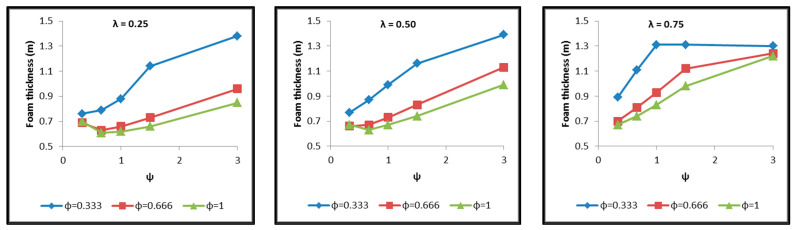
Distribution of the required insulating material thickness for maintaining the boil-off rate at 0.1%/h as a function of the ratios λ, φ and ψ.

**Figure 19 materials-16-02245-f019:**
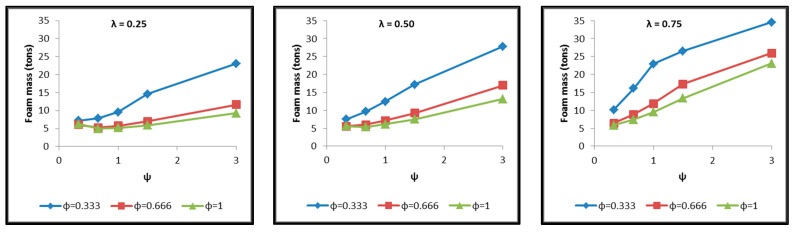
Distribution of the required insulating material mass for maintaining the boil-off rate at 0.1%/h as a function of the ratios λ, φ and ψ.

**Figure 20 materials-16-02245-f020:**
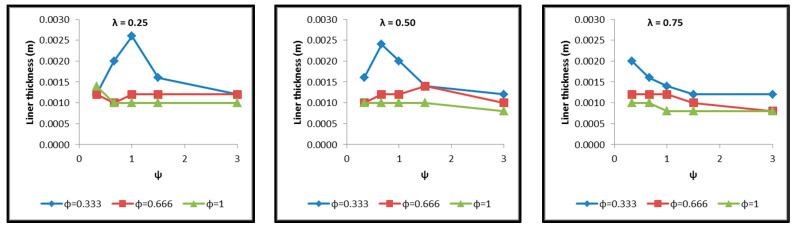
Distribution of inner aluminum shell thickness for evaluation of the Von Mises criterion as a function of the ratios λ, φ and ψ.

**Figure 21 materials-16-02245-f021:**
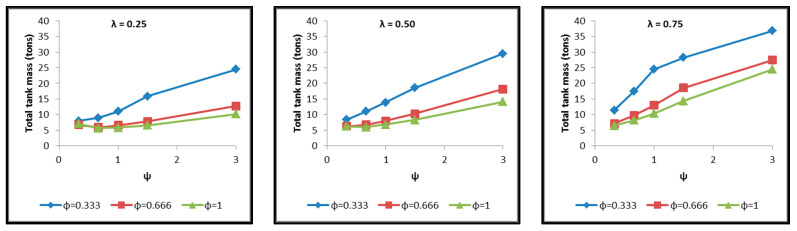
Distribution of total tank mass as a function of the ratios λ, φ and ψ.

**Figure 22 materials-16-02245-f022:**
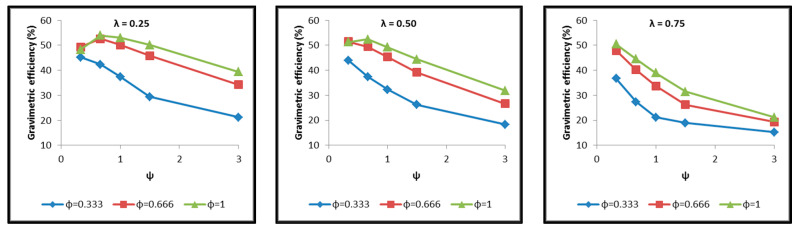
Distribution of the tank gravimetric efficiency as a function of the ratios λ, φ and ψ.

**Figure 23 materials-16-02245-f023:**
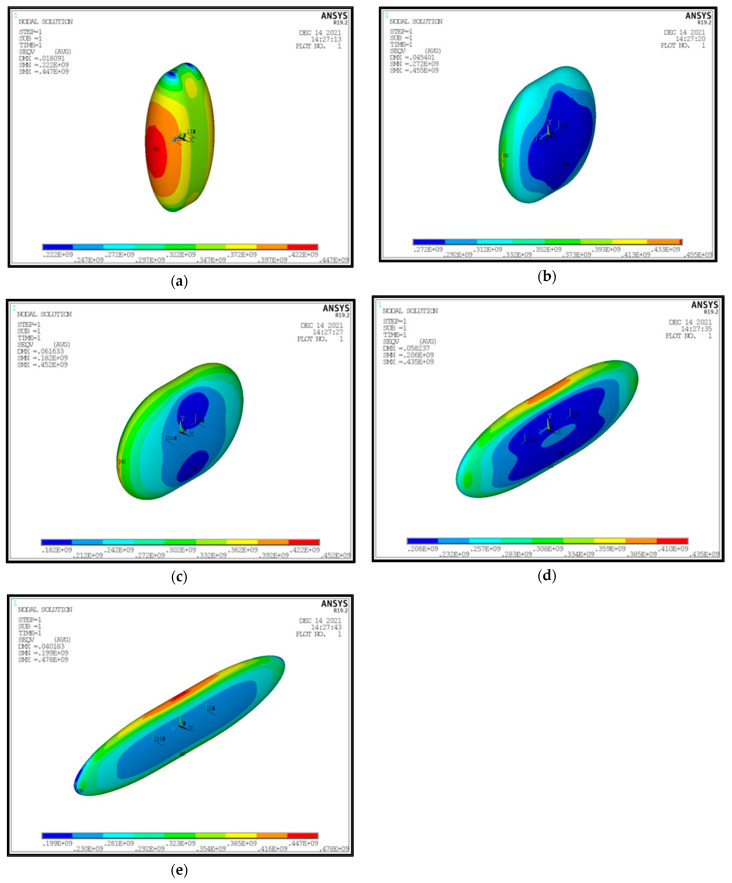
Characteristic tank case with transition of the maximum Von Mises stress between different tank regions (λ = 0.25, φ = 0.333, ψ = 0.333 (**a**), 0.666 (**b**), 1 (**c**), 1.5 (**d**), 3 (**e**)).

**Table 1 materials-16-02245-t001:** LH_2_ properties compared to Jet-A (kerosene) fuel [4,5,6].

Property	LH_2_	Jet-A (Kerosene)
Energy content per unit mass (MJ/kg)	120	42.8
Energy content per unit volume (MJ/L)	8.49	31.15
Density (kg/m^3^)	71	811
Specific heat capacity (J/gK)	9.69	1.98
Specific heat of vaporization (J/kg)	461,000	360,000

**Table 2 materials-16-02245-t002:** Properties of the composite material of the outer tank shell.

Property	Value
Density, ρ	1600 kg/m^3^
Thermal conductivity, k	k_11_ = 3.972 W/mK, k_22_ = k_33_ = 0.3363 W/mK
Specific heat capacity, C_p_	1000 kg/m^3^
Coefficient of thermal expansion, a_ii_	a_11_ = −0.2 × 10^−6^ m/m/°K, α_22_ = α_33_ = 30 × 10^−6^ m/m/°K
Young’s modulus, E_ii_	E_11_ = 133 GPa, Ε_22_ = Ε_33_ = 9.13 GPa
Poisson vatio, v_ij_	v_12_ = v_13_ = 0.3, v_23_ ≅ 0.3
Shear modulus, G_ij_	G_12_ = G_13_ = 5.2 GPa, G_23_ ≅ 5.2 GPa
Ply longitudinal tensile strength, X_t_	2178.2 MPa
Ply longitudinal compressive strength, Y_t_	91.7 MPa
Ply transvers tensile strength, X_c_	1783.5 MPa
Ply transverse compressive strength, Y_c_	340.9 MPa
Ply in-plane shear strength, S	129.1 MPa

**Table 3 materials-16-02245-t003:** Output results for the selection of the mesh density.

Elements by Thickness	Element Size in Plane	Total Number of Elements	Q (W)	T_min_ (K)	T_max_ (K)
4	0.15 m	32,208	942.4	23.56	282.47
4	0.10 m	75,888	943.9	23.56	282.47
4	0.05 m	289,872	945.1	23.56	282.47
6	0.15 m	42,944	923.6	23.48	282.48
6	0.10 m	101,184	923.8	23.48	282.48
6	0.05 m	386,496	924.5	23.48	282.48
8	0.15 m	53,680	916.9	23.46	282.48
8	0.10 m	126,480	916.4	23.45	282.48
8	0.05 m	483,120	916.8	23.45	282.48

**Table 4 materials-16-02245-t004:** Geometric characteristics of the tank configurations studied under constant volume.

λ	φ	ψ	l_t_ (m)	l_s_ (m)	a (m)	b (m)	c (m)
0.25	0.333	0.333	4.649	1.162	1.743	1.743	5.236
		0.666	7.380	1.845	1.384	2.768	4.156
		1	9.677	2.419	1.208	3.629	3.629
		1.5	12.681	3.170	1.056	4.755	3.170
		3	20.130	5.032	0.838	7.549	2.516
	0.666	0.333	3.690	0.923	2.768	1.384	4.156
		0.666	5.858	1.464	2.197	2.197	3.298
		1	7.681	1.920	1.918	2.880	2.880
		1.5	10.065	2.516	1.676	3.774	2.516
		3	15.977	3.994	1.330	5.991	1.997
	1	0.333	3.223	0.806	3.629	1.208	3.629
		0.666	5.116	1.279	2.880	1.918	2.880
		1	6.708	1.677	2.515	2.515	2.515
		1.5	8.790	2.197	2.197	3.296	2.197
		3	13.953	3.488	1.744	5.232	1.744
0.5	0.333	0.333	5.882	2.941	1.471	1.471	4.416
		0.666	9.337	4.669	1.167	2.334	3.505
		1	12.243	6.122	1.019	3.061	3.061
		1.5	16.043	8.022	0.890	4.011	2.674
		3	25.467	12.734	0.707	6.367	2.122
	0.666	0.333	4.669	2.334	2.334	1.167	3.505
		0.666	7.411	3.705	1.853	1.853	2.782
		1	9.718	4.859	1.618	2.429	2.429
		1.5	12.734	6.367	1.413	3.183	2.122
		3	20.213	10.107	1.122	5.053	1.684
	1	0.333	4.077	2.039	3.061	1.019	3.061
		0.666	6.472	3.236	2.429	1.618	2.429
		1	8.486	4.243	2.122	2.122	2.122
		1.5	11.120	5.560	1.853	2.780	1.853
		3	17.652	8.826	1.471	4.413	1.471
0.75	0.333	0.333	9.045	6.784	1.131	1.131	3.395
		0.666	14.358	10.769	0.897	1.795	2.695
		1	18.827	14.121	0.784	2.353	2.353
		1.5	24.671	18.503	0.685	3.084	2.056
		3	39.163	29.372	0.543	4.895	1.632
	0.666	0.333	7.179	5.384	1.795	0.897	2.695
		0.666	11.396	8.547	1.425	1.425	2.139
		1	14.943	11.208	1.244	1.868	1.868
		1.5	19.581	14.686	1.087	2.448	1.632
		3	31.083	23.313	0.863	3.885	1.295
	1	0.333	6.270	4.702	2.353	0.784	2.353
		0.666	9.952	7.464	1.868	1.244	1.868
		1	13.050	9.787	1.631	1.631	1.631
		1.5	17.100	12.825	1.425	2.138	1.425
		3	27.145	20.359	1.131	3.393	1.131

## Data Availability

Not applicable.

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
