# Peer review of "Computational Analysis of Liquid Hydrogen Storage Tanks for Aircraft Applications"

_materials, 2023, doi:10.3390/ma16062245_

Round 1

Reviewer 1 Report

The authors presented the analysis of useful research topic related to hydrogen for fuel in aircraft under cryogenic temperature scnerieos. The tanks are considered as working domain and then the dynamics parametrically analyzed via an efficient scheme known as Finite Element Method. Then under various parametric ranges, the detailed analysis provided. The manuscript is written well and the results are interesting and these would be beneficial for aerodynamics discipline. However, the moderate changes required before the acceptance of the article. 

1. The manuscript should be checked for grammatical mistakes and run spell check throughout.

2. What research gaps the authors found after performing the literature review? These should be added in the last paragraph of the introductions  section.

 3. The materials and method section must contains the assumptions and the problem statement clearly presented because this is very important section that help to reader for understanding the problem.

4. The quality of Fig. 2 should be improved and is not clear in the present form.

5. How your work support and use for practical applications further? 

6. Which type of governing equations used for the model formulation? These are missing and should be included in the revised version.

7. Conclusions section needs improvement with core findings of the study.

Author Response

Manuscript ID: materials-2216957

Title: Computational analysis of liquid hydrogen storage tanks for aircraft applications

By: Vasileios K. Mantzaroudis and Efstathios E. Theotokoglou

Journal: Materials

Thorough revision of the work has been performed by the authors in order to address the constructive input of the reviewers and improve the final quality of the manuscript. The comments of the reviewers have been carefully addressed. Also, in the revised version of the paper, special attention has been paid to correct various typographical and syntax errors.

Response to comments of Reviewer #1

The authors would like to thank the Reviewer for his insightful comments and suggestions.

  1. The manuscript has been checked for grammatical mistakes and run spell check throughout

 2 From the literature it appears that there is a lack of computational methods for dealing with tanks of different geometry configuration and LH2 level store. This is mentioned in the last paragraph of the Introduction at Section 1 of our revised version

 3 The assumptions are fully described in Section 2 of our revised version. For the better understanding these are also introduced in the first paragraph of Section 2.

 4 The quality of Fig 2 has been improved

5 The use of new tanks from new materials and different geometries has a great influence in new airplane structures. In addition liquid hydrogen tanks except of the aviation industry may be deployed at the maritime transport industries, at the industrial chemistry, etc. These are mentioned in the last paragraph of the Conclusions’ section 4 of our revised version.

6 Our study focuses on the global computational analyses of liquid hydrogen storage tanks based on the assumptions presented in section 2 examining the effect of the LH2 level stored, as well as the tank geometric configurations. In particular, the proposed study concerns a global thermal and static computational analysis for parameterized simplistic tank geometries; the case of dynamic analysis and the introduction of governing equations related to a specific geometry with large or small curvatures will be the purpose of a next study. These are mentioned in the last paragraph of the Conclusions section 4 of our revised paper.

  1. The conclusions section 4 has been improved with the core findings of our study. The core findings are mentioned in the last paragraph of the Conclusions section 4 of our revised paper.

Reviewer 2 Report

In this work, Mantzaroudis et al. conducted computational analysis of a liquid hydrogen tank, which could be used in aircraft. This work is conducted in good format, however, certain aspects of this work could be improved. Comments attached below.

1.     Typo/wording/formality

Typos such as “an challenging”, “170 kPapressure” should be taken care of 

2.      Edge cases

To be used in aircraft, the tank will go through extreme conditions, such as taking off and landing, where the liquid hydrogen should not be studied in a static state. May the authors consider this situation as well?

Author Response

Manuscript ID: materials-2216957

Title: Computational analysis of liquid hydrogen storage tanks for aircraft applications

By: Vasileios K. Mantzaroudis and Efstathios E. Theotokoglou

Journal: Materials

Thorough revision of the work has been performed by the authors in order to address the constructive input of the reviewers and improve the final quality of the manuscript. The comments of the reviewers have been carefully addressed. Also, in the revised version of the paper, special attention has been paid to correct various typographical and syntax errors.

Response to comments of Reviewer #2

The authors would like to thank the Reviewer for his insightful comments and suggestions

  1. We have corrected all the appeared typos.
  2. In the present study we confront the static problem considering a global computational analysis and focusing on the material selection, on the geometry of the tank and on the LH2 level store. The dynamic case that is also a very interesting case will be the study of a future work. These are mentioned in the last paragraph of the Conclusions section 4 of our revised paper.

Reviewer 3 Report

This is well-written and timely work. However feel to ask a few questions such as in terms of the tank, are they fixed or there is any possibility to change the dimension and properties? If so how your model can cope with this?

Also wondering to make this suitable for materials mdpi authors should use more reference from that journal or at least from mdpi. 

 https://doi.org/10.3390/ma16041655

https://doi.org/10.3390/aerospace9120801

Figure 23 : ANSYS analysis results caption are hardly seen hence the different colour and its indications are not understandable.  Improve this. 

Figure 18-22 good piece of information

Figure 9-11 hard to understand

where can this liquid hydrogen tank be deployed except in the aviation industry.

If possible include one abbreviation or nomenclature section to indicate the detail of all the notation.

Author Response

Manuscript ID: materials-2216957

Title: Computational analysis of liquid hydrogen storage tanks for aircraft applications

By: Vasileios K. Mantzaroudis and Efstathios E. Theotokoglou

Journal: Materials

Thorough revision of the work has been performed by the authors in order to address the constructive input of the reviewers and improve the final quality of the manuscript. The comments of the reviewers have been carefully addressed. Also, in the revised version of the paper, special attention has been paid to correct various typographical and syntax errors.

Response to comments of Reviewer #3

The authors would like to thank the Reviewer for his insightful comments and suggestions.

1) Our analysis constitutes the case of analyzing computationally proper tanks to examine the effect of the LH2 level stored, as well as the tank geometric configurations. Thus in our study we cope with the change of dimensions. The problem of changing the material properties will be the scope of a future research.

2) According to the suggestion of the Reviewer we have included in our reference list of our revised version, 2 more papers from mdpi journals ([34, 35]) and we mentioned them in our text.

3) We have improved Fig. 23.

4) We have improved Figures 9-11.

5) Liquid Hydrogen tank except of the aviation industry may be deployed at the maritime transport industries, in industrial chemistry, etc. These are mentioned in the last paragraph of the Conclusions section 4 of our revised paper.

6) Because the notations of various magnitudes are included in the text of our revised version we didn’t added a separate nomenclature section.

Round 2

Reviewer 3 Report

accept